# Comparison of Thermophilic–Mesophilic and Mesophilic–Thermophilic Two-Phase High-Solid Sludge Anaerobic Digestion at Different Inoculation Proportions: Digestion Performance and Microbial Diversity

**DOI:** 10.3390/microorganisms11102409

**Published:** 2023-09-27

**Authors:** Tianfeng Wang, Jie Wang, Jiajia Pu, Chengxiang Bai, Cheng Peng, Hailong Shi, Ruoyu Wu, Ziying Xu, Yuqian Zhang, Dan Luo, Linhai Yang, Qingfang Zhang

**Affiliations:** College of Petrochemical Engineering, Lanzhou University of Technology, Lanzhou 730050, China; 18624464182@163.com (J.W.); P3543261547@163.com (J.P.); bcx18438598789@163.com (C.B.); pengc_ee@163.com (C.P.); q1974400490@163.com (H.S.); wuruoyu924@163.com (R.W.); xzykiyo999@163.com (Z.X.); 13463125121@163.com (Y.Z.); luodan@lut.edu.cn (D.L.); yanglh_lz@163.com (L.Y.); zhangqf@lut.edu.cn (Q.Z.)

**Keywords:** ammonia, inhibition, methane production, bacteria community, archaea community

## Abstract

This study investigated the performance of thermophilic–mesophilic (T-M) and mesophilic–thermophilic (M-T) two-phase sludge anaerobic digestion at different inoculation proportions after a change in digestion temperature. After temperature change, the pH, total ammonia nitrogen (TAN), free ammonia nitrogen (FAN), solubility chemical oxygen demand (SCOD), and total alkalinity (TA) levels of two-phase digesters were between thermophilic control digesters and mesophilic control digesters. However, the volatile fatty acid (VFA) levels of two-phase digesters were higher than those of thermophilic or mesophilic control digesters. The bacteria communities of M-T two-phase digesters were more diverse than those of T-M. After a change in digestion temperature, the bacterial community was dominated by *Coprothermobacter*. After a change of digestion temperature, the relative abundance (RA) of *Methanobacterium*, *Methanosaeta*, and *Methanospirillum* of M-T two-phase digesters was higher than that of T-M two-phase digesters. In comparison, the RA of *Methanosarcina* of T-M two-phase digesters was higher than that of M-T two-phase digesters. The ultimate methane yields of thermophilic control digesters were greater than those of mesophilic control digesters. Nevertheless, the ultimate methane yield levels of M-T two-phase digesters were greater than those of T-M two-phase digesters. The ultimate methane yields of all two-phase digesters presented an earlier increase and later decrease trend with the increasing inoculation proportion. Optimal methane production condition was achieved when 15% of sludge (T-M15) was inoculated under mesophilic–thermophilic conditions, which promoted 123.6% (based on mesophilic control) or 27.4% (based on thermophilic control). An optimal inoculation proportion (about 15%) balanced the number and activity of methanogens of high-solid sludge anaerobic digestion.

## 1. Introduction

Municipal waste-activated sludge is an inevitable byproduct of municipal sewage treatment plants. Over 60 million metric tons of sludge (80% water content) are produced annually in China [1]. Sludge is rich in nutrients for plant growth (nitrogen, phosphorus, potassium, etc.) but also contains a certain amount of readily fermentable organic matter, pathogenic bacteria, trace amounts of organic matter, and emerging pollutants [1,2]. Before further utilization or disposal, sludge needs to undergo special treatment to enhance biostability, inactivate pathogenic bacteria, and reduce emerging pollutants [3]. Anaerobic digestion (AD) is a feasible technology for sludge treatment, which results in biostability enhancement, pathogenic bacteria inactivation, emerging pollutants reduction, and biogas methane generation [4,5].

The AD process involves hydrolysis, acidogenesis, acetogenesis, and methanogenesis [6]. The hydrolysis stage of anaerobic digestion is the first stage, designed to convert organic waste into soluble organic matter by hydrolytic bacteria [6]. The acidogenesis and acetogenesis stages of anaerobic digestion are mainly characterized by the production of volatile fatty acids, especially acetic acid, via acid-forming bacteria [7]. During the methanogenic stage, acetic acid, H_2_, and CO_2_ are converted into biogas via the synergistic action of different methanogens [8]. The four stages above depend on the synergistic effect of a complex microflora [8].

As to conventional sludge AD, the AD process is performed in a single reactor system [6]. However, for hydrolytic bacteria, acid-forming bacteria, and methanogens, the environmental requirement, thermodynamic properties, and kinetic constants are discrepant, which results in low overall AD efficiency [9]. To enhance the overall AD efficiency of anaerobic digestion, “two-phase digestion” is used to separate acid-producing and methanogenic phases [10]. Two-phase AD is required to enable the production of organic acids (acidification stage) and methane (methanogenesis step) to have their suitable environmental conditions, which helps to improve the stability and efficiency of the whole AD process [11].

According to the digester temperature, two-phase AD systems mainly fall into thermophilic–mesophilic (T-M), mesophilic–thermophilic (M-T), thermophilic–thermophilic (T-T), mesophilic–mesophilic (M-T), hyperthermophilic–thermophilic (H-T), and hyperthermophilic–mesophilic (H-M) [8]. Combined with the VS removal, methane production, energy balance, retention time, and system stability, T-M and M-T are reasonable options for two-phase sludge AD [7,11].

Wang et al. (2018) compared T-M two-phase, thermophilic single-phase, and mesophilic single-phase high-solid sludge AD [12]. The relative abundance (RA) of Methanosarcina from high to low was T-M two-phase, thermophilic single-phase, and mesophilic single-phase. Compared with single-phase high-solid sludge AD, the total methane production improved by 81.6% (M-T/mesophilic) or 74.8% (M-T/thermophilic). Chen and Chang (2020) compared the T-M and M-T two-phase sludge AD [13]. The acidification phase affects the methanogenic degree by changing methanogen composition and enhancing the system stability by enriching *Methanosarcina*. Compared with single-phase sludge AD (mesophilic or thermophilic), the total methane production improved by 23.1% (T-M/mesophilic) and 11.9% (M-T/thermophilic). He et al. (2023) investigated the effect of conventional and intermittent temperatures on methanogenic performance, and microbial regulation of anaerobic fermentation was investigated. The results showed that the hydrolysis rate increased with the increase in temperature [14]. The intermittent temperature significantly increased methane production by 39.4% and 19.0% compared to the mesophilic and thermophilic digesters. Compared with single-phase anaerobic digestion, two-phase anaerobic digestion can indeed effectively improve the anaerobic digestion performance of sludge; it has a good improvement effect on methane yield, volatile solid (VS) removal, and solubility chemical oxygen demand (SCOD) removal. During the two-phase anaerobic digestion, the improvement of biogas production and removal of organic pollutants depends on their stability and performance of the hydrolyzed acidic and methanogenic phases. In addition, the hydrolytic acidic phase is a rate-limiting step, and it was found that two-phase anaerobic digestion can improve anaerobic digestion performance by improving hydrolysis [14].

However, the difference between T-M and M-T two-phase high-solid sludge AD is still being determined. High ammonia of high-solid sludge AD is likely to influence the composition of methanogens; anaerobic digestion of high-solid sludge also affects hydrolysis, mass transfer, dewatering performance, and inhibitor concentration; temperature also affects hydrolysis, gas production, and sludge viscosity, and the interplay of many influences can lead to complex results [1]. The performance of the inoculated sludge also significantly influences the anaerobic digestion of sludge, affecting the growth of microorganisms and the rate of methane production. Moreover, the proportion of inoculation is a crucial factor in two-phase AD, which influences the microbial community, substrate concentration, inhibitor concentration, and methane production of AD [15,16,17].

Therefore, this study investigated the effects of five different inoculation proportions on T-M and M-T two-phase high-solid sludge AD. The difference in methane production and microbial community was observed during the two-phase high-solid sludge AD process.

## 2. Materials and Methods

### 2.1. Materials

Waste-activated sludge and dewatered sludge were acquired from a local municipal sewage treatment plant in Lanzhou, China. The plant’s capacity was 100,000 m^3^·d^−1^, using an anaerobic–anoxic–oxic process. Waste-activated sludge anaerobic digestion under thermophilic or mesophilic conditions for 20 days was used as inoculum. Table 1 shows the characteristics of initial inoculums and substrate.

### 2.2. Experimental Setup

Glass bottles described by Liu et al. (2022) were used as anaerobic digesters. Dewatered sludge inoculated with thermophilic or mesophilic inoculum at a mixing ratio of 4:1 (*w*/*w*, wet basis) [18]. The mass of the substrate (dewatered sludge and inoculum) was 750 ± 1 g. To ensure an anaerobic environment, all digesters were flushed with high-purity nitrogen (>99.99%) gas over 3 min to remove the remaining air. Then, all digesters were put in the water bath kettle at thermophilic (55 ± 2 °C) or mesophilic (37 ± 2 °C) temperature.

On the 11th day, a portion of the digesters were taken apart. The corresponding digested sludge was used as thermophilic or mesophilic inoculum for two-phase AD. Different proportions of thermophilic or mesophilic inoculum were inoculated into mesophilic or thermophilic digesters. Five different mesophilic digested sludge inoculation proportions (mesophilic inoculum/substrate = 0%, 5%, 10%, 15%, and 20%. wet basis) were labeled as T-M0, T-M5, T-M10, T-M15, and T-M20 of T-M two-phase AD. Five different thermophilic digested sludge inoculation proportions (thermophilic inoculum/substrate = 0%, 5%, 10%, 15%, and 20%. wet basis) were labeled as M-T0, M-T5, M-T10, M-T15, and M-T20 of M-T two-phase AD. The thermophilic and mesophilic digesters of the whole digestion process were labeled as T-C and M-C, respectively. Each working condition was performed in triplicate.

### 2.3. Analytical Methods

The pH, TSS, VSS, TS, and VS of sludge samples were determined using the standard method [18]. The total ammonia nitrogen (TAN) and total alkalinity (TA) of liquid samples were determined using the standard method [19]. The free ammonia nitrogen (FAN) proportion in TAN was calculated via a special computational formula determined by the pH and temperature [19]. As to soluble chemical oxygen demand (SCOD), volatile fatty acids (VFAs), methane content of biogas, and biogas volume, the determination method was the same as described by Liu et al. (2022) [18]. We generally use a Hash SCOD meter to test for SCOD. Relative abundance (RA) is a metric used to represent the proportion of each category in a set of data.

The sludge samples’ deoxyribonucleic acid (DNA) extraction method was the same as described by Liu et al. (2022) [18]. Finally, the raw data and species information were processed on the website (http://www.ncbi.nlm.nih.gov, accessed on 1 October 2022) to generate a complete table of operational taxonomic units (OTUs). The reference thoroughly describes the experimental procedures and data preprocessing techniques [20].

### 2.4. Statistical Analysis

Statistical analysis was used to determine the correlation using SPSS software Version 22.0. Both correlations and variations were statistically significant at a confidence interval of *p* < 0.05.

## 3. Results and Discussion

### 3.1. pH, TAN, and FAN

The pH of the anaerobic digester is mainly related to the balance of alkalinity and VFA [21]. The pH levels at the thermophilic temperature were higher than those at the mesophilic temperature (Figure 1a,b). This phenomenon should be related to the increment of alkalinity. The sludge protein deamination generated ammonia at different levels (Figure 1c,d), which resulted in different alkalinity levels [22]. The pH of all digesters was between 7.2 and 8.8, which was a reasonable pH range for microbial metabolism of anaerobic digestion [23]. The pH levels of T-M0, T-M5, T-M10, T-M15, and T-M20 were close and remained stable after a change of digestion temperature. This phenomenon should be related to the different hydrolysis abilities between thermophilic and mesophilic temperatures [24,25]. Low hydrolysis ability at mesophilic temperature makes it hard to hydrolyze organic matter, which has been hydrolyzed at thermophilic temperature. Nevertheless, high hydrolysis ability at thermophilic temperature can effectively hydrolyze organic matter that has been hydrolyzed at mesophilic temperature [7]. The high pH levels in this experiment compared to the relevant literature may be because the change in the anaerobic digestion rate of sludge after a temperature change can lead to a change in pH levels [26]. During anaerobic digestion, acid-producing bacteria break down organic matter into volatile fatty acids (VFA) such as acetic, propionic, and butyric acids. These acids combine with alkali metal ions in the reactor to produce the corresponding salts. The conversion of VFA to methane via methanogenic bacteria consumes hydrogen ions and may result in an increase in pH [27].

TAN is the hydrolysis product of proteins [18], which increase gradually with the development of the digestion process (Figure 1c,d). A high concentration of TAN or FAN has an inhibitory effect on microorganisms [28]. There is a dynamic equilibrium between TAN and FAN in the system, which is influenced by temperature and pH [7]. The TAN levels of T-M0, T-M5, T-M10, T-M15, and T-M20 were slightly lower than that of T-C, similar to M-C at a later stage of the digestion process. The TAN levels of M-T0, M-T5, M-T10, M-T15, and M-T20 were significantly higher than that of M-C after a change in digestion temperature. The abovementioned phenomenon should be related to the fact that thermodynamic properties and kinetic constants at thermophilic temperature are greater than those at mesophilic temperature [7,11]. The differences in ammonia nitrogen between this study and other related studies are insignificant [1,3,26], and many factors affect ammonia nitrogen. Temperature greatly influences the anaerobic digestion process and microbial activity [26]. Microbial activity is important in removing ammonia nitrogen during anaerobic digestion [3]. Factors such as insufficient nutrients and accumulation of toxic substances may affect microbial activity, which in turn affects the removal of TAN [28]. Excessive ammonia nitrogen load may lead to ammonia nitrogen accumulation and affect the treatment effect. Reasonable control of the influent ammonia nitrogen load is essential to maintain the stable operation of the anaerobic digestion system [8].

During anaerobic digestion, FAN is also toxic to methanogens [2]. The FAN of T-C is higher than that of M-C (Figure 1e,f). The FAN of T-M0, T-M5, T-M10, T-M15, and T-M20 was similar. It remained the same after the temperature change (about 400 mg L^−1^). The FAN of M-T0, M-T5, M-T10, M-T15, and M-T20 was similar, which increased quickly after a change of digestion temperature (from 200 mg L^−1^ to 1600 mg L^−1^). This phenomenon should be related to the synchronous increase of pH and TAN. The FAN at a thermophilic temperature was significantly greater than that at a mesophilic temperature. The slightly higher levels of FAN in our study compared to the other literature may be due to the effect of pH, temperature, and nutrient imbalance.

Higher or lower may lead to weakened microbial activity and affect the removal of TAN, while higher temperatures may lead to the accumulation of volatile fatty acids and affect the methanogenesis process [29]. Microbial activity plays an important role in removing ammonia nitrogen during anaerobic digestion. Factors such as insufficient nutrients and accumulation of toxic substances may affect microbial activity, which in turn affects the removal of TAN. Excessive ammonia nitrogen load may lead to ammonia nitrogen accumulation and affect the treatment effect. Reasonable control of the influent ammonia nitrogen load is essential to maintain the stable operation of the anaerobic digestion system [8]. A high pH results in elevated free ammonia concentrations [2]. Generally, anaerobic digestion processes are best served in the pH range of 6.5–7.5. When the pH is lower than 7, free ammonia tends to accumulate; when the pH is higher than 7.5, free ammonia may be released into the environment as ammonia gas, thus affecting the effectiveness of anaerobic digestion. Temperature has a great influence on microbial activity during anaerobic digestion. When the temperature is higher or lower, the microbial activity may be affected, leading to the accumulation of free ammonia [30]. During anaerobic digestion, microorganisms require sufficient nutrients to remain active. An imbalance of nutrients such as nitrogen and phosphorus in the system may lead to a decrease in microbial activity, affecting the removal of free ammonia [31].

### 3.2. VFA, SCOD, TA, and VFA/TA

VFA (Figure 2) are the products of hydrolysis and acidification and are also substrates for biogas production [22]. As to day 0, day 4, and day 8, the VFA levels of M-C were higher than that of T-C. This should be related to a high methane production rate of T-C at the initial stage of AD, which resulted in low VFA. As to day 11, day 19, day 27, day 33, and day 49, the VFA gradually decreased. After a change of digestion temperature (thermophilic to mesophilic/mesophilic to thermophilic), the VFA levels significantly increased. Moreover, the VFA levels of the mesophilic–thermophilic system were higher than those of the thermophilic–mesophilic system. Overall, the VFA levels at the thermophilic temperature were greater than those at the mesophilic temperature. This phenomenon is similar to the change of TAN. Both VFA and TAN are representative hydrolysis products that have a positive correlation with temperature [18]. The VFA content in this study did not differ significantly from the VFA content in the other related literature [3,32]. There are many factors affecting VFA, such as organic loading, substrate inhomogeneity, etc. [2]. Excessive organic loading may lead to VFA accumulation. When the organic load is higher, the decomposition of organic substances in the anaerobic digestion system is faster, which may lead to VFA accumulation [7]. Therefore, it is necessary to control the organic load reasonably according to the actual conditions. Poor agitation and mixing may lead to localized areas of VFA accumulation in anaerobic digestion systems [8]. Therefore, maintaining good agitation and mixing is essential to ensure homogeneous reaction conditions [33].

The main component of SCOD is VFA [1]. The SCOD was significantly positively correlated to VFA (Figure 3a,b). All the digesters showed a trend of first rising and then falling. The SCOD of T-C was significantly higher than that of M-C. Meanwhile, the SCOD levels of two-phase anaerobic digesters were between T-C and M-C. This phenomenon should be related to hydrolysis ability at a thermophilic temperature being greater than that at a mesophilic temperature [7,11]. The pH value has a great influence on the microbial activity during anaerobic digestion. Usually, the anaerobic digestion process is optimal in the pH range of 6.5–7.5. When the pH is lower than 7, SCOD is easy to accumulate; when the pH is higher than 7.5, SCOD may be released into the environment in the form of gas, thus affecting the anaerobic digestion effect [34].

TA (Figure 3c,d) is considered a crucial factor in anaerobic digestion, but its role is to ensure that the anaerobic system has a specific buffering capacity and maintains a suitable pH [32]; once the anaerobic system is acidified, it takes a long time to recover. The TA of T-C was higher than that of M-C, which might be caused by the complete degradation of organic matter in a thermophilic temperature than in a mesophilic temperature [18]. After the temperature change, the TA levels of T-M0, T-M5, T-M10, T-M15, and T-M20 were close, which showed a general trend of gentle increase. The TA levels of M-T0, M-T5, M-T10, M-T15, and M-T20 were similar, showing a growth trend followed by a gentle increase after digestion temperature change. This phenomenon may be due to the degradation of organic matter in the digestive system as digestion proceeds, leading to a rise in the TA of the system [32]. In this study, the levels of alkalinity may be slightly higher than in the other literature, probably because of pH and microbial activity [35]. TA is closely related to the pH of the wastewater. In the anaerobic digestion process, when the pH value is greater than 7.5, the TA is easy to accumulate, thus affecting the anaerobic digestion effect [35]. TA has a significant effect on microbial activity. During anaerobic digestion, the balance between organic acids and alkalis produced by microorganisms significantly affects total alkalinity. When microbial activity decreases, total alkalinity may be affected [36].

TVFA/TA can be used as an early warning indicator for system operation. The critical value of TVFA/TA is 0.35 [7]. When the TVFA/TA is greater than this value, the system may be acidified and destabilized [37]. The TVFA/TA (Figure 3c,d) showed that all digesters were stable. The levels of TVFA/TA in this paper do not differ much from the other literature, and the reactor gradually becomes stabilized after a period of time from the beginning of the reaction [38]. During anaerobic digestion, the equilibrium relationship between organic acids and alkaline substances produced by microorganisms significantly affects the TVFA/TA ratio [3]. When the microbial activity decreases, the TVFA/TA ratio may be affected [39].

### 3.3. Methane Generation

The cumulative methane yields of T-C were higher than that of M-C (Figure 4a,b), probably because the hydrolytic bacteria, acid-forming bacteria, and methanogens in the thermophilic temperature digesters were more active than those in the mesophilic temperature digesters [40].

The cumulative methane yields of T-M0, T-M5, T-M10, T-M15, and T-M20 were similar, which is close to the cumulative methane yields of T-C. This phenomenon should be related to the low hydrolysis ability at mesophilic temperature, reflected by similar VFA and SCOD levels after a temperature change.

The cumulative methane yields of M-T0, M-T5, M-T10, M-T15, and M-T20 were greater than that of M-C (Table 2). With the increasing inoculation proportion, cumulative methane yields presented an earlier increase and later decrease trend. The highest cumulative methane yield was 133.47 mL g^−1^ VS_added_ (M-T15). Compared with T-C or M-C, the cumulative methane yields increased 31.65% or 123.61%. The volumetric methane production rate (Figure 4d) increased with the increase of the inoculation proportion at the initial stage after a temperature change, which resulted from different numbers of methanogens. Meanwhile, the FAN also increased with the increased inoculation proportion, which resulted from the high TAN of T-C on the 11th day. High FAN resulted in ammonia inhibition [6]. Therefore, there was a balance between the number of methanogens (positive correlation with inoculation proportion) and the activity of methanogens (negative correlation with inoculation proportion). The methanogenic performance of T-M in this study was slightly lower compared to the other literature, and the methanogenic performance of M-T was improved compared to the other literature, probably due to the effect of temperature, oxygen reduction potential, and nutrients [41]. Temperature is a key factor affecting anaerobic digestion performance for methane production. Usually, the optimal temperature range in the anaerobic digestion process is 35–38 °C. In this temperature range, the microbial activity is higher, and the methane production performance is better. The pH value has a great influence on the microbial activity during anaerobic digestion. Usually, the anaerobic digestion process is optimal in the pH range of 6.5–7.5. When the pH is lower than 7, methanogenic performance may be affected; methanogenesis may be inhibited when the pH is higher than 7.5 [42,43]. The redox potential reflects the electron transfer conditions in an anaerobic digestion system. In the appropriate redox range (usually between −200 and −400 mV), microbial activity is higher, and methanogenic performance is better [44].

### 3.4. Bacteria Distribution

On the ninth day, at mesophilic temperature, the bacterial community before the temperature change was dominated by *Terrimonas* (91.93%) and *Coprothermobacter* (4.17%) (Figure 5). At thermophilic temperature, the bacterial community was dominated by *Coprothermobacter* (99.04%) (Figure 5). *Terrimonas* is a Gram-negative bacterium belonging to the phylum Bacteroidetes, an important phylum in mesophilic anaerobic digesters associated with the breakdown of proteins and cellulose [45]. Bacteroidetes, colonies such as *Terrimonas*, can break down simpler organic substances such as fatty acids and lactic acid, producing hydrogen and some organic acids [41]. *Coprothermobacter* is mainly involved in the acid-producing phase of anaerobic digestion [46]. *Coprothermobacter* has strong activity at thermophilic temperatures, and it can cooperate with hydrotropic methanogenic bacteria to degrade organic pollutants [47]. *Coprothermobacter* and *Methothermobacter* can act synergistically to degrade organic matter [48,49,50]. On the 27th day, the bacteria communities of M-T0, M-T5, M-T10, M-T15, and M-T20 were more diverse than that of T-M0, T-M5, T-M10, T-M15, and T-M20. This phenomenon fits perfectly with the results of VFA and SCOD, which was an important reason for the high methane yield of M-T two-phase digesters. On the 43rd day, all digesters were dominated by *Coprothermobacter* (except M-C). The RA of *Terrimonas* gradually decreases with the progress of the digestion process. This phenomenon should be related to the fact that proteins and cellulose have been greatly broken down in the early stage of the digestion process [30,45].

### 3.5. Archaea Distribution

On the ninth day, the archaea community (Figure 6) of M-C was dominated by *Methanobacterium* (49.74%), *Methanosaeta* (29.99%), *Methanothermobacter* (12.11%), *Methanosarcina* (2.91%), and *Methanospirillum* (1.62%). The archaea community of T-C was dominated by *Methanothermobacter* (53.93%), *Methanosarcina* (32.60%), *Methanobacterium* (6.00%), *Methanospirillum* (1.85%), and *Methanosaeta* (1.55%). *Methanothermobacter* is a thermophilic hydrogenotrophic methanogenic archaeon capable of reducing CO_2_ with H_2_ to produce methane [51]. *Methanothermobacter* can grow in different temperature ranges, which makes them model organisms for studying thermophiles and organisms in extreme environments [52]. Therefore, the RA of *Methanothermobacter* at a thermophilic temperature was higher than that at a mesophilic temperature. Moreover, *Methanothermobacter* is also very effective in degrading proteins [52], which is beneficial to sludge anaerobic digestion (proteins are the main component of sludge organic matter). *Methanosarcina* typically grows at temperatures in the range of 30–45 °C, with some strains able to grow at a wider range of temperatures [27]. *Methanosarcina* is an acetic acid-nutrient methanogenic archaea, which means they utilize acetic acid as a substrate for growth and methanogenesis [47]. Therefore, the RA of *Methanosarcina* at mesophilic temperature was higher than that at thermophilic temperature. Moreover, *Methanobacterium* (hydrogenotrophic) and *Methanospirillum* (acetotrophic), as methanogens, are synergistic during the anaerobic digestion process and can interact with other methanogens and fermenters to promote the decomposition and conversion of organic matter [53]. *Methanosaeta* typically grows at temperatures in the range of 30–45 °C, with some strains being able to grow at a wider range of temperatures [37]. *Methanosaeta* is a complex methanogen that can utilize various organic substances (e.g., formic acid, methanol, formaldehyde, etc.) as substrates for growth and methanogenesis [27]. The abovementioned properties make *Methanosaeta* highly adaptable to anaerobic environments [39].

On the 27th day, the archaea communities of M-C were dominated by *Methanosaeta* (50.06%), *Methanospirillum* (23.34%), *Methanobacterium* (9.31%), *Methanothermobacter* (4.98%), and *Methanosarcina* (2.00%). The archaea community of T-C was dominated by *Methanothermobacter* (49.77%), *Methanosarcina* (34.03%), *Methanobacterium* (5.96%), *Methanosaeta* (3.42%), and *Methanospirillum* (3.00%). The archaeal community of mesophilic–thermophilic two-phase digesters was dominated by *Methanothermobacter* (37.10–55.96%), *Methanosarcina* (12.65–21.79%), *Methanobacterium* (17.18–21.46%), *Methanosaeta* (3.43–12.95%), and *Methanospirillum* (3.00–9.86%). The archaeal community of thermophilic–mesophilic two-phase digesters was dominated by *Methanothermobacter* (26.38–53.28%), *Methanosarcina* (32.08–57.56%), *Methanosaeta* (1.50–7.88%), *Methanobacterium* (2.54–5.89%), and *Methanospirillum* (1.56–4.23%).

On the 43rd day, the archaea community of M-C was dominated by *Methanosaeta* (50.59%), *Methanospirillum* (20.62%), *Methanobacterium* (8.61%), *Methanothermobacter* (3.67%), and *Methanosarcina* (1.91%). The archaea community of T-C was dominated by *Methanothermobacter* (55.02%), *Methanosarcina* (26.71%), *Methanobacterium* (5.16%), *Methanosaeta* (3.61%), and *Methanospirillum* (1.71%). The archaeal community of mesophilic–thermophilic two-phase digesters was dominated by *Methanothermobacter* (21.54–43.95%), *Methanosarcina* (14.76–36.57%), *Methanobacterium* (15.76–25.53%), *Methanosaeta* (5.74–11.44%), and *Methanospirillum* (3.23–8.37%). The archaeal community of thermophilic–mesophilic two-phase digesters was dominated by *Methanothermobacter* (30.95–47.18%), *Methanosarcina* (32.35–57.75%), *Methanosaeta* (2.07–5.05%), *Methanobacterium* (1.60–6.01%), and *Methanospirillum* (0.92–1.68%).

The archaea distribution on the 27th and the 43rd days was similar. The RA of *Methanobacterium*, *Methanosaeta*, and *Methanospirillum* of M-T two-phase digesters was higher than that of T-M two-phase digesters. The RA of *Methanosarcina* of T-M two-phase digesters was higher than that of M-T two-phase digesters. The adaptability to the different substrates of *Methanobacterium*, *Methanosaeta*, and *Methanospirillum* was higher than that of *Methanosarcina* [39,47,53]. The abovementioned phenomenon should be an important reason for the higher methane yield of M-T two-phase digesters than that of T-M two-phase digesters.

## 4. Conclusions

The bacteria communities of M-T two-phase digesters were more diverse than those of T-M. The ultimate methane yield levels of M-T two-phase digesters were greater than those of T-M two-phase digesters. The ultimate methane yields of all two-phase digesters presented an earlier increase and later decrease trend with the increasing inoculation proportion. An optimal inoculation proportion (about 15%) for methane production balanced the number and activity of methanogens of high-solid sludge anaerobic digestion.

## Figures and Tables

**Figure 1 microorganisms-11-02409-f001:**
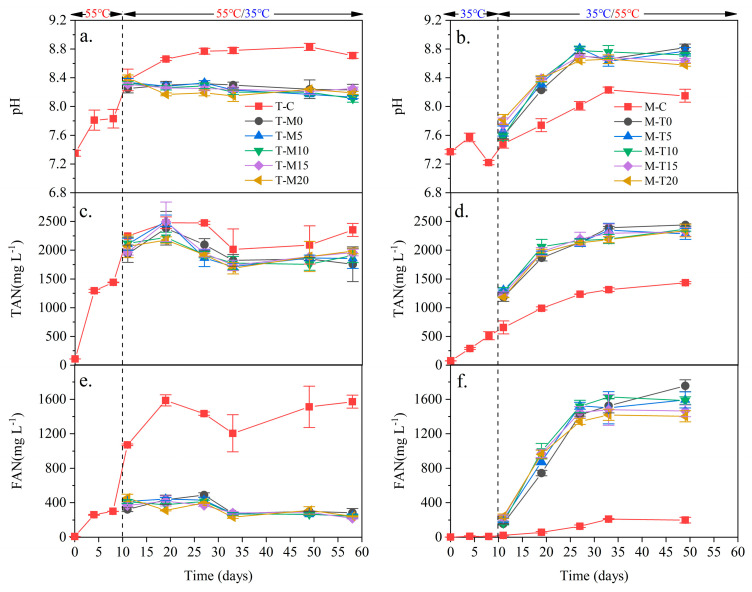
Changes in pH, TAN, and FAN during the digestion process ((**a**) pH of T-M; (**b**) pH of T-M; (**c**) TAN of T-M; (**d**) TAN of M-T; (**e**) FAN of T-M; and (**f**) FAN of M-T).

**Figure 2 microorganisms-11-02409-f002:**
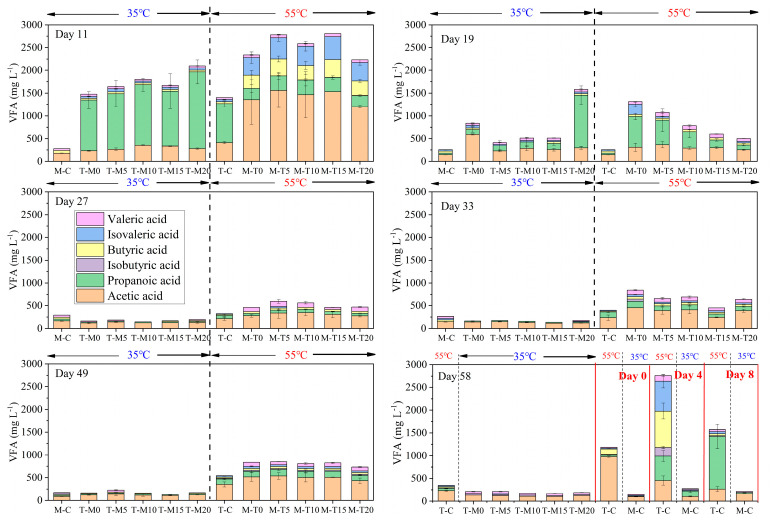
Changes of VFA during the digestion process (M: 55 ± 2 °C; T: 35 ± 2 °C).

**Figure 3 microorganisms-11-02409-f003:**
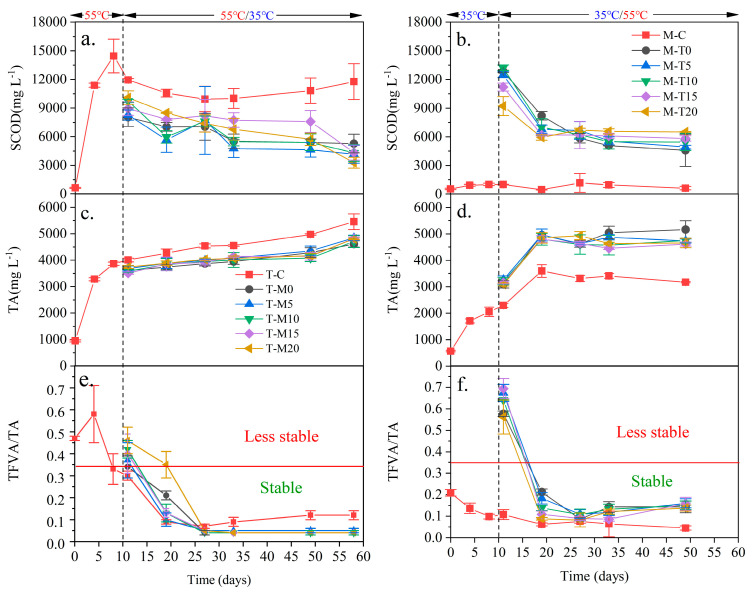
Changes of SCOD, TA, and VFA/TA during the digestion process ((**a**) SCOD of T-M; (**b**) SCOD of T-M; (**c**) TA of T-M; (**d**) TA of M-T; (**e**) TVFA/TA of T-M; and (**f**) TVFA/TA of M-T).

**Figure 4 microorganisms-11-02409-f004:**
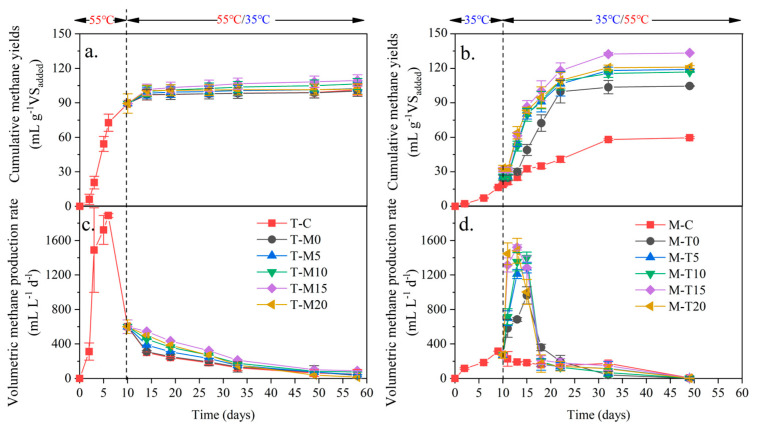
Changes in methane generation during the digestion process ((**a**) Cumulative methane yields of T-M; (**b**) cumulative methane yields of T-M; (**c**) volumetric production rate of T-M; and (**d**) volumetric production rate of M-T).

**Figure 5 microorganisms-11-02409-f005:**
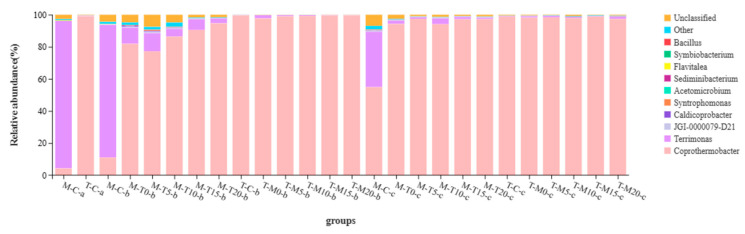
Relative abundance of bacterial communities at the genus levels (a: on 9th day; b: 27th day; c: 43rd day).

**Figure 6 microorganisms-11-02409-f006:**
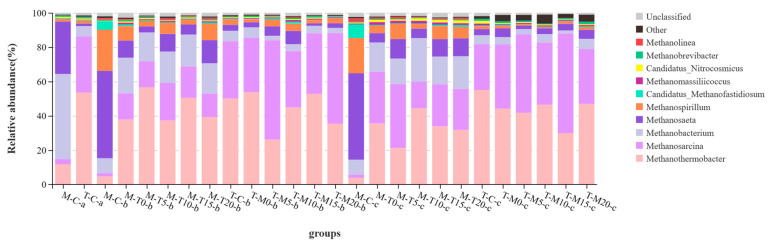
Relative abundance of archaea communities at the genus levels (a: on 9th day; b: 27th day; c: 43rd day).

**Table 1 microorganisms-11-02409-t001:** Characteristics of initial inoculums and substrate (VSS: volatile solids; TSS: total solids).

	Mesophilic Inoculum	Thermophilic Inoculum	Dewatered Sludge
pH	7.48 ± 0.21	7.69 ± 0.15	
VSS (g L^−1^)	9.3 ± 0.1	8.0 ± 0.2	
TSS (g L^−1^)	16.8 ± 0.1	14.8 ± 0.3	
TS (%)	1.7 ± 0.0	1.5 ± 0.0	16.2 ± 0.1
VS/TS (%)	56.1 ± 0.6	54.1 ± 1.3	44.3 ± 0.5
carbon (dry basis) (%)			21.84 ± 0.12
nitrogen (dry basis) (%)			3.94 ± 0.13

Note: All values are expressed as mean ± standard deviation (*n* = 3).

**Table 2 microorganisms-11-02409-t002:** Parameters of the modified Gompertz model.

	Ultimate Methane Yields, UMYs (mL g^−l^ VS_added_)	Maximum Methane Production Rate, R_m_ (mL g^−1^ VS_added_ d^−1^)	Lag Phase, λ (d)	R^2^
T-C	98.07	18.48	1.91	0.997
T-M0	97.88	18.53	1.94	0.998
T-M5	99.85	17.81	1.87	0.997
T-M10	102.51	17.59	1.86	0.995
T-M15	105.08	17.03	1.81	0.993
T-M20	100.71	18.06	1.91	0.997
M-C	61.42	2.50	2.86	0.991
M-T0	108.46	7.30	7.65	0.983
M-T5	118.34	10.60	7.84	0.993
M-T10	116.96	10.87	7.98	0.994
M-T15	132.87	11.15	7.57	0.995
M-T20	120.29	18.06	1.91	0.994

## Data Availability

Not applicable.

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
