# Peer review of "Comparison of Thermophilic–Mesophilic and Mesophilic–Thermophilic Two-Phase High-Solid Sludge Anaerobic Digestion at Different Inoculation Proportions: Digestion Performance and Microbial Diversity"

_microorganisms, 2023, doi:10.3390/microorganisms11102409_

Round 1

Reviewer 1 Report

The authors performed a comprehensive study on thermophilic-mesophilic and mesophilic-thermophilic two-phase sludge anaerobic digestion at different inoculation ratios. The quality of the conducted research is good, however, the manuscript needs to be modified accordingly before being considered for publication.

Comments:

1-The manuscript needs to be proofread and revised by a native speaker. There are grammatical errors throughout the manuscript. 

2-Table 1, why TS% VS%,etc. are not provided for the mesophilic and thermophilic inocula?

3-Figure 1, it could be confusing for the readers to understand the conditions for each figure. Please better specify in the caption.

4-Figure 2, please specify the AD temperature.

5-Figure 3 and Figure 4, see the comment for figure 1.

6-The main findings of the present research need to be better specified in the abstract. 

The manuscript needs to be revised by a native speaker. There are grammatical errors throughout the manuscript. 

Author Response

Please see the document.

Reviewer 2 Report

Comments on the manuscript titled: Comparison of thermophilic-mesophilic and mesophilic-thermophilic two-phase high-solid sludge anaerobic digestion at different inoculation proportion: digestion performance and microbial diversity by Wang et al. submitted to microorganisms-mdpi

General Comments:

The reviewed manuscript on comparative thermophilic-mesophilic and mesophilic-thermophilic two-phase studies is very interesting. However, I would like to address some inquiries to the authors and ask them to improve some technical issues before possible publication.

It would be advised to target the manuscript to a wider audience than just the Chinese community. Therefore, it would be useful to cite references regarding municipal sewage more generally in the World.

In lines 32 and 36 Authors mention emerging contaminants. According to definition of this term these usually refer to many different kinds of chemicals, including medicines, personal care or household cleaning products, lawn care and agricultural products, among others. The types of emerging contaminants which have been found in groundwater worldwide include: caffeine and nicotine and their metabolites, flame/fire retardants and surfactants, industrial additives and by-products, nanomaterials (very small particles) as well as personal care products and fragrances. The question is how this is connected to the presented research?

Although the title suggests microbial diversity, only scarce information about the results concerning this issue is present in Abstract. Please specify.

Make sure that all the abbreviations are explained just after they appear in the text for the first time. 

Materials and Methods:

Lines 101-102 activated sludge anaerobic digested under thermophilic or mesophilic conditions for 20 days? Was it really 20 days? Or 10? Like indicated on figures

Table 1 please add a footnote or a header explaining what the used abbreviations mean concerning characteristics of initial inoculums and substrate. Please specif how it was measured and how many measurements were executed to calculate an erros (SD?)

Please add the details how the Gompertz model was created in Materials and Methods. 

Results and discussion:

Make sure that the characteristics listed in subchapters are previously explained .

Figure 2. It would be more convenient to translate these data into a table and sort it according to time; start from day 0, than4, 8 etc. When certain acids are within a columns or lines it will be easier to compare the data.

SCOD is not explained in Materials and Methods.

Lines 257-258 please give references concerning “other literature”

Line 266 explain why value of 0,35 was choosen. If ref [7] explains that please move the reference further.

Technical issues:

1. Since the authors represent the same scientific unit, there is no need to number it. A mere * next to the name of the corresponding author is sufficient.W abstrakcie nie mogÄ… znajdować siÄ™ żadne niewyjaÅ›nione skróty jak TAN, FAN, SCOD, TA etc.

2.      Please pay attention to abbreviations. An explanation of the abbreviation should appear when it first appears in the text.

3.      Sections of text in some lines have different formatting.

3.      It is not necessary to create a census of all authors from ref 21. Please check microorganisms-mdpi guidelines

4.      Of the 60 references, 34 are from China. Is this type of research really done mainly in China? Please correct me and make the paper more internationalLine 107 Liu et al. (2022) please add reference numer [??];  the same in line 134

5.      Insert spaces between a characteristics and units in Fgures

6.      Line 172 higher “than that of M-C”

7.      SCOD, RA and other characteristics not explained in Materials and Methods

some minor corrections are required

Author Response

Please see the document.
